# Smart Mixes in International Supply Chains: A Definition and Analytical Tool, Illustrated with the Example of Organic Imports into Switzerland

**Robert Home ***, **Mareike Weiner and Christian Schader**

Department of Socioeconomics, Research Institute of Organic Agriculture, Ackerstrasse 113,
5070 Frick, Switzerland; mareike.weiner@fibl.org (M.W.); christian.schader@fibl.org (C.S.)
* Correspondence: robert.home@fibl.org

**Abstract:** Combinations of national and international, hard and soft powers, known as Smart Mixes, have been proposed as a way for governments in consumer countries to influence the behaviour of supply chain actors who operate outside their jurisdiction. However, the Smart Mix concept has not yet been precisely defined, which has hindered its operationalization as a means of analyzing the governance of long and complex international supply chains. The aim of this contribution is to derive a working definition of Smart Mixes and use it to create and demonstrate a generalizable analytical tool that facilitates identification of whether a Smart Mix exists in an international supply chain. To address this aim, we reviewed existing literature on Smart Mixes to define the concept, which led to a three-step process for analyzing a supply chain. In a second stage, we demonstrate the process by applying it to the example of organic imports into Switzerland, using data from expert interviews and existing public documentation. The application showed that the governance of the organic sector in Switzerland related to imported products fulfils the criteria for it to be considered a Smart Mix that enables the Swiss Government to influence the behaviour of supply chain actors outside its jurisdiction. This example shows that the proposed Smart Mix concept is sound under particular circumstances: in this case, when the interests of the public and private sectors are aligned so that binding public measures provide protection to the private sector. These circumstances are not unique to organic imports into Switzerland, which allows the conclusion that Smart Mixes may provide sustainability solutions in other international supply chains.

**Keywords:** smart mix; sustainability; organic; international supply chain; voluntary measures; Switzerland

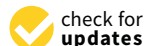



## 1. Introduction

Environmental and social aspects of sustainability are gaining importance in many economic sectors as both the environmental limits of growth and blatant human rights violations become increasingly apparent (European Commission 2014; Labowitz and Baumann-Pauly 2015; Steffen et al. 2015). Governments use command and control measures to alter the commercial arrangement between sub-suppliers, suppliers, and buyers to promote adoption and long-term use of more sustainable business practices (Hartmann and Moeller 2014). However, governments in consumer countries do not have the power to legislate for behavioral change by actors outside their jurisdiction, which limits their ability to ensure the sustainable production of products in international supply chains, which are often long and complex. This calls for rethinking how governments in consumer countries can apply their power to ensure that the supply chains of goods consumed within their countries meet acceptable environmental and social sustainability standards.

Wettstein (2015, p. 6) points out that "neither voluntary market-based approaches nor a grand legal framework on their own" can create change but, in combination, could generate incremental and cumulative change to achieve their goals. These combinations have come to be known as "Smart Mixes," with the term first used by Gunningham

et al. (1998) and mainstreamed by Ruggie (2011) who also added the requirement to include voluntary measures. However, the logic of Smart Mixes has been challenged, with Kinderman (2016, p. 31) literally calling for a "reality check" by arguing that "business organizations want to keep private governance private and that in most cases, public authorities have to overcome business opposition" to establish arrangements in which businesses adopt sustainable supply practices. Cossart et al. (2017) similarly noted that government efforts to force corporate social responsibility (CSR) are usually met with strong opposition from the business sector, although are usually accepted if they are part of a business-friendly package tax deal. Examples of this strategy include Mauritius (Kinderman 2016), the U.K. (Kinderman 2012), and India (Van Zile 2011), in which the CSR clauses were added to make tax cuts for business more palatable to the public.

Kinderman (2016, p. 30) stated that "a smart mix implies that private governance and hard law regulation are complementary or at the very least compatible. The suggestion seems to be that a Smart Mix combines the best of both worlds: the flexibility, dynamism, innovativeness, reflexivity and adaptability of voluntary market-based solutions and the authoritativeness, scope, and binding force of legal regulation." Ruggie's (2011) idea and conception of a Smart Mix includes an expectation that legislation would follow from voluntary efforts by industry (Wettstein 2015). Kinderman (2016) continues by challenging the political viability of the Smart Mix concept and argues against the notion that business organizations will voluntarily participate in Smart Mixes simply because they are needed.

An explanation for the discrepancy, with some scholars proposing Smart Mixes as a solution, while others challenge the logic behind them, might be that the description of Smart Mixes in the UN guiding principles (UNGPs) "as an intelligent mix of national and international, binding and voluntary measures" (Ruggie 2011) leaves considerable room for interpretation. In other words, scholarship into the viability of the Smart Mix concept has been hampered by the lack of agreement on a working definition. Kinderman's (2016) critique of the underlying logic of Smart Mixes may therefore be the result of a different understanding of which "measures" should be included, what characteristics allow a collection of measures to be considered a "mix," and what conditions allow a mix to be considered "intelligent."

The aim of this study is to derive a clear working definition of a Smart Mix, which we argue is necessary for the concept of Smart Mixes to be operationalized. A definition would enable the development of a process, or tool, with which the smartness of combinations of sustainability measures in existing supply chains can be analyzed and potential improvements to the mix can be diagnosed. A second aim is to develop and demonstrate such a process. To illustrate the definition, we use the example of certification of organic produce imported into Switzerland in which the private sector adopts market-based measures to promote environmental or social sustainability. Specifically, we examine whether the mix of measures that can be found in the Swiss organic sector and which are relevant to imports indicate a convergence between the interests of public and private actors' interests (Kinderman 2016) and interact to improve sustainability outcomes in the parts of the sector outside the jurisdiction of the Swiss government.

We begin by reviewing relevant literature to derive a definition of a Smart Mix along with providing a clear understanding of the terms that make up the definition. From this definition, we operationalize the concept by describing a process to analyze international supply chains, with a focus on identifying which measures exist and how they interact. Finally, we demonstrate the process by applying it to organic products imported into Switzerland to evaluate whether the existing public and private measures within the Swiss organic sector meet the criteria for them to be considered a Smart Mix. By way of conclusion, we draw some recommendations for possible additions to the existing Smart Mix and offer some suggestions for future research into the operationalization of the concept.

## 2. Literature Review

### 2.1. Governance in Consumer Countries for Sustainable Supply Chains

Governance has been conceptualized as "the regulation and coordination of activities by public and private institutions through a variety of formal and informal instruments" (Boström et al. 2015, p. 2). Sustainability regulations, such as national laws regulating work safety, environmental damage, and social security, took decades to operationalize legally in the aftermath of the social, environmental, cultural, and economic turmoil of industrialization (Salminen 2019). Prior to this operationalization, emphasis was on freedom of contract and property as primary organizational principles behind regulating new concentrations of labor, supply, distribution, and consumer contracts within jurisdictions (Salminen and Rajavuori 2019). Voluntary corporate social responsibility, if it anachronistically may be called so, is one answer to the ensuing social and environmental problems (Sydow et al. 2021). This debate showed that national sustainability laws are much more complex than simple command and control style regulation, with various stakeholders included in, and affecting, legislation and its development in diverse ways (Salminen 2019).

Governments in consumer countries therefore attempt to align framework conditions, by motivating public support for desirable behavior, so that private actors freely and voluntarily act in the interests of society and contribute to ecological and social goals. This means that sustainable behavior at company level also has to be in the interest of shareholders. However, McMurtry (1997) points out that arguments for a free market are only coherent if there is systematic omission of non-business costs and benefits, which are seen as the responsibility of the state, and that such an omission fails to comprehend their economic value. Kinderman (2016, p. 39) put it bluntly that "existing scholarship tends to over-state the convergence of public and private actors' interests."

Chandler (2006, p. 66) explains the motivations of the private sector by stating that "The whole of corporate history shows unequivocally that protection of the interests of stakeholders other than shareholders has not come from voluntary corporate measure, but from extended pressure followed by legislation. This has been true of labor conditions, of protection of the environment and, today, observance of human rights. The market economy survives today because it is not a 'free market,' but one bounded by moral parameters enforced by law." The strongest motivations for corporations engaging in environmental behavior has been found to be compliance with regulations, followed by cutting costs (Hendry and Vesilind 2005). Kinderman (2016) further notes the fallacy of any assumption that regulation is unnecessary and that corporations might voluntarily act in the interests of stakeholders other than shareholders. Furthermore, Kinderman (2016) agrees that corporate sustainability measures in the fields of labor conditions, environment, and human rights have rarely come from voluntary measures, but rather from the extended pressure followed by the legislation that Chandler (2006) suggests. These aspects play a crucial role for companies in the management and optimization of their business practices and their value chains as well as in communication with customers and end consumers (Eccles et al. 2014). Chandler (2006, p. 66) continues: "the challenge is to extend those parameters to embrace the values of contemporary society without diminishing the dynamism which needs to underlie the corporate contribution."

### 2.2. Jurisdictional Constraints in International Supply Chains

However, there remains a tendency for environmental damage (Godar et al. 2016) and human rights violations (Salmivaara 2018) to be concentrated in exporting countries, which makes the Western consumer an unwilling contributor to unsustainable social and environmental practices in the global south (Hassel and Helmerich 2016). Governments in producer countries are limited in their ability to legislate for social and environmental sustainability because of the risk of losing the industry, and the jobs that come with it, to a competing host country with a higher degree of relative attractiveness (Gimet et al. 2015). The concentration of unsustainable practices in producer countries means that governments of consumer countries are faced with the challenge of how to apply their

power to promote sustainable supply chains when the majority of the chain, and most of the problems, lie outside their jurisdiction.

Apart from the move away from such ideas of "command and control" regulation in legal scholarship, as exemplified in Amstutz and Karavas (2009), jurisdictional limits would seem to be a question of developments in legal operability, as noted for example by Salminen and Rajavuori (2019). Recent proposals, such as the German administration's plans towards a broader and more stringent version of an earlier French law (BMAS 2021), seek to overcome the challenge of operationalizing transnational sustainability regulations (Salminen and Rajavuori 2019). The political trajectory of many such planned instruments is still uncertain, as witnessed by the watering down of an originally more advanced Swiss Responsible Business Initiative to a "fig-leaf-like counter-proposal" (SCCJ 2021), although the tendency towards regulation is clear and seemingly more rapid than it was in relation to national sustainability laws in the aftermath of industrialization. This more diverse picture of the regulatory space in relation to extraterritorial effects is becoming increasingly politically palatable, as evidenced by the European Parliament's (2021) request for an EU-level transnational sustainability law, and better understood from a legal operative perspective. However, attempts to translate "soft" law into "hard" law by converting non-binding features of the UNGPs into obligations for ratifying States have faced opposition from a number of States (Trebilcock 2020).

In principle, the governments of consumer countries face the challenge of convincing others "to act in ways in which that entity would not have acted otherwise" by applying the hard and soft powers at their disposal (Wilson 2008). In practice, they must seek ways to motivate private industry to undertake particular behaviors voluntarily that might commonly be understood as being in the public interest. The UN guiding principles (UNGPs) offer a solution by calling upon states "to consider an intelligent mix of national and international, binding [hard power] and voluntary [soft power] measures in order to promote respect for human rights by companies" (Ruggie 2011). Although the UNGPs were referring to social sustainability, their logic appears to apply equally to issues of environmental sustainability. Van Erp et al. (2019, p. 5) concur with the observation that "policy instruments taken separately (for example, liability rules, taxation, and emission trading or command and control regulation) have particular limitations, thus justifying a need for a combination of instruments." Ruggie's (2011) idea and conception of a Smart Mix includes an expectation that legislation would follow from voluntary efforts by industry (Wettstein 2015). This is in line with Kinderman (2016, p. 29), who describes Smart Mixes as "the mix of voluntary and regulatory measures, which has emerged as a compromise formula between advocates of voluntary, private, market-based regulation and advocates of hard law."

### 2.3. Defining a Smart Mix

Gunningham et al.'s (1998) initial description of a Smart Mix allowed the term to apply to a variety of mixes of actors, levels of governance, or institutional structures with optional private measures. The common usage has since evolved to contain a requirement for a voluntary component. Ruggie (2011, p. 5) declared that "States should not assume that businesses invariably prefer, or benefit from, State inaction, and they should consider a Smart Mix of measures—national and international, mandatory and voluntary—to foster business respect for human rights." The implication is that a Smart Mix is a mix of actions taken by a state in which policy can combine binding and non-binding measures to contribute to the effective and efficient achievement of environmental and social goals (Swiss Federation 2016). Rees (2019, n.p.) goes as far as to say that a Smart Mix must "comprise measures across all those categories of 'national and international' and 'mandatory and voluntary' [...] A truly 'Smart Mix' means looking at all four aspects [...] not just the one or two that are most convenient or already in place. [...] The UNGPs also clearly contemplate mandatory international measures as a natural part of this 'Smart Mix.'"

However, the distinction between public and private measures is not always clear, with a number of measures being voluntary with public participation, for example, the OECD's Guidelines for Multinational Enterprises, which are government-backed recommendations on responsible business conduct to encourage sustainable and responsible business conduct (OECD 2019). Although they are legally non-binding, the OECD Investment Committee and its Working Party on Responsible Business Conduct encourage implementation and the 49 adhering countries are required to set up a National Contact Point, which are offices charged with promoting observance of the Guidelines by multinational enterprises. Despite government involvement, and some obligations, they would earlier have been categorized as international/voluntary. To cater for such measures, we suggest replacing the public/private distinction in previous definitions of Smart Mixes with a three-tiered public/hybrid/private categorization consisting of "binding public," "voluntary cooperative," and "voluntary private."

Rees' (2019, n.p.) requirement to include both national and international aspects also leaves some questions unanswered. We argue that the significance of "international" is that it implies being outside the jurisdiction of the regulating body so that the relevant sustainability issue cannot be simply solved with legislation. However, there are circumstances in which there might be jurisdictional demarcation within a single country, such as when a national government wants to interfere with issues that are governed by states. We therefore suggest relaxing the requirement of "international" and replace it with a requirement that a Smart Mix must have consequences outside the jurisdiction of the intervening government, which will include an international aspect in most cases.

A further implication is that a mix will be "smart" if there is some alignment of policy with the measure(s) that a business undertakes voluntarily. Kinderman (2016, p. 29) underlines this point by stating that "a Smart Mix implies that private governance and hard law regulation are complementary or at the very least compatible," while Van Erp et al. (2019) define smart as meaning that the mix addresses both the cause and solution to the problem it is intended to solve. Van Erp et al. (2019), however, caution that governance arrangements are path dependent with context-specific impacts that depend on the specific institutional, social, economic, and environmental conditions, particularly in cases of polycentric governance, which is normally the case given the "international" requirement.

Combinations of institutions and actors emerge spontaneously and interact, often in unexpected and unintended ways (Van Erp et al. 2019), which challenges the idea that Smart Mixes can be purposefully designed by a central actor. Governments overcome this challenge by creating guidelines or standards to give legitimacy to voluntary measures, which should motivate businesses to undertake what these guidance measures suggest by changing to more sustainable practices. Governments therefore have a key role to play in shaping the interaction between public and private standards (von Hagen and Alvarez 2012). Integrating private governance systems, such as certifications and consumer labels, can therefore produce even more effective and efficient combinations of measures, for example, to reduce transaction costs on the government side (Rørstad et al. 2007; UN Working Group on Business and Human Rights 2014).

Van Erp et al. (2019, p. 50) synthesized the descriptions of Smart Mixes to define them as the mix of "various regulatory and governance instruments, both public and private and both international and local, [which] can be combined into sophisticated mixes of complementary instruments and actors, tailored to the specific needs of the situation." Van Erp et al.'s (2019) definition of a Smart Mix can be broken down into specific criteria for identifying real-world examples of Smart Mixes but neglects both the interactions and the performance of the specific measures. That the measures should interact is a straightforward requirement for a mix to be "smart," but performance is less clear. Van Erp et al.'s (2019) requirement that the Smart Mix should address an intended problem suggests that performance of a Smart Mix relates to the capacity of these measures to mutually enhance achievement of stated objectives, rather than focusing on the overall achievement of a sustainable supply chain. Furthermore, there is no implication that the objectives of

the component measures of a Smart Mix must be aligned or agreed, but rather that the interactions between measures must simply enhance the achievement of objectives of one of the measures. A summary of the evolution of the characteristics of Smart Mixes is shown in Table 1.

**Table 1.** Evolution of the definition of Smart Mix.

| Author | Characteristics |
| --- | --- |
| Gunningham et al. (1998) | Combination of actors, levels of governance or institutional structures. Business interaction is optional. |
| Ruggie (2011) | Combination of measures undertaken by a state: National and international, mandatory and voluntary. Businesses interaction is implied. |
| Swiss Federation (2016) | Combination of measures undertaken by a state: mandatory and voluntary. Guidelines to achieve environmental and social goals are necessary. |
| Kinderman (2016) | Combination of measures undertaken by a state: National and international, mandatory and voluntary. Business interaction is necessary. Guidelines to achieve environmental and social goals are implied. |
| Van Erp et al. (2019) | Combination of measures undertaken by a state: National and international, mandatory and voluntary. Business interaction is necessary. Guidelines to achieve environmental and social goals are necessary. |

Synthesizing these definitions enables formulation of the working definition for Smart Mix for the purposes of this study:

*A Smart Mix is a combination of measures that includes at least one binding public measure, accompanied by at least one voluntary cooperative measure that gives guidance to the actions that should be undertaken to achieve stated objectives and at least one voluntary private measure that must have consequences outside the jurisdiction of the intervening government. The combination of measures must interact and thus improve the achievement of the objectives of at least one of the measures.*

### 2.4. Clarifying Key Terms

Before the definition of a Smart Mix can be operationalized, it is necessary to clarify what is meant by the key terms. As there is no uniform, generally accepted classification of policy instruments in the literature (Bemelmans-Videc et al. 2011), definitions of the three types of measures that form a Smart Mix are first needed.

#### 2.4.1. Binding Public Measures

Firmly within the coercive power of governments is the application of what Nye (2004) classified as hard powers. "Hard" legally binding measures can create clarity about expected behavior of organizations or individuals, which facilitates the identification of non-compliant behavior (European Commission 2017). Hard powers can be divided into coercive force by means of command-and-control regulation and coercion by the use of market-based/economic instruments. Within the national jurisdiction, coercive force is typically the traditional governance through legislative measures, which can be defined as "the direct regulation of an industry or activity by legislation that states what is permitted and what is illegal" (Junquera and Del Brío 2016, p. 1). However, regulation commonly provides no financial incentive for going beyond limits, offers limited flexibility on how to achieve goals, is economically less efficient, and is commonly susceptible to politically motivated loopholes (Junquera and Del Brío 2016). Furthermore, legislative hard powers are limited in their scope to create desired outcomes outside the jurisdiction of the legislating body.

Governments also have the power to regulate behavior within their jurisdiction, which can have effects outside it. Due diligence legislation essentially outsources the responsibility for controlling "rather than imposing pre-ordained behavioural rules" (Bartley 2014, p. 94) and holds companies "morally, politically and legally accountable for their activities, or those of their suppliers, abroad" (Partzsch and Vlaskamp 2016, p. 978). Mandatory due diligence usually obliges companies to ensure that the materials they use have been sourced in compliance with a nominated set of guidelines, such as the 2010 EU Timber Regulation (EUTR) in the timber sector. The EUTR requires importers who place timber products on the EU market for the first time to undertake a risk management exercise so as to minimize the risk of placing illegally harvested timber, or timber products containing illegally harvested timber, on the European Single Market (European Parliament 2010). Due diligence legislation essentially requires companies to "conduct an ongoing, proactive, and reactive checking process in their supply chain in order to identify and manage the risk of contributing, directly or indirectly, to social and/or environmental harm" (Partzsch and Vlaskamp 2016, p. 978).

However, the 'due diligence' approach is subject to some weaknesses and can imply a substantial investment if done seriously (Trebilcock 2020), so companies may actively attempt to avoid their responsibilities—especially in complex supply chains in which traceability is difficult (Koch and Kinsbergen 2018). A further weakness of approaches based on mandatory due diligence is that individual importers may lack the capacity to effectively control the sourcing of their materials. They are thereby forced to discharge their due diligence obligations by outsourcing the controlling, such as to certification bodies, which may be unreliable. This weakness was highlighted by the 2019 timber scandal in which, despite the EUTR and certification by the green label, Program for the Endorsement of Forest Certification (PEFC), 100,000 tons of illegally sourced timber was imported into the EU from Russia (Harding 2020). Despite these weaknesses, due diligence legislation is seen as a powerful measure for supply chain sustainability: especially when supported by laws coupled with effective home state sanctions, such as administrative fines, on the relevant lead firm, as for example, the U.S. Foreign Corrupt Practices Act and the U.K. Bribery Act. Indeed, in the 2019 case of Lungowe v Vedanta, the U.K. Supreme Court decided that lead firms are liable for the inadequate governance of their extra-jurisdictional value chains (Norton Rose Fulbright LLP 2019).

Market-based instruments within the national jurisdiction include (i) incentives, which are inducements, or supplemental rewards that serve as motivations for a desired action or behavior and (ii) taxes, which can be imposed to punish undesired behavior. An advantage that market-based instruments have over legislative instruments is their economic efficiency (Bishop 1993). Trade arrangements are a key measure for improving social and environmental standards in global value chains (Aissi et al. 2017; Myant 2017). The EU has promoted labor standards in several recent trade agreements and the approach is particularly used by the U.S. While trade benefits, such as exemption from, or reduction of, duties are provided, the respect of the desired social or environmental standards by the counterpart can be required. These requirements often address the state level, such as to undertake legal and institutional changes or to strengthen trade union rights (Aissi et al. 2017). The U.S. model of market-based coercion in foreign trade agreements is described as "conditional" and links compliance with a desired behavior to trade privilege (Harrison et al. 2019). In contrast, the approach typically taken by the EU is described as "promotional" and provides a framework for dialogue and cooperation rather than linking compliance to incentives (Harrison et al. 2019).

### 2.4.2. Voluntary Cooperative Measures

Soft powers are those for which there are no penalties for non-compliance and can be broadly defined as when a government can motivate another actor to want what it wants through attraction rather than coercive force or market-based coercion (Nye 2004). It arises from the attractiveness of the motivator's culture, political ideals, or policies (Nye 2004).

Inherent in this concept is that the wishes of the motivating government are communicated, which can take a variety of forms such as guidelines, standards, and certification schemes. Soft instruments are more flexible approaches that can be implemented by the government without any other actor contributing. However, soft powers are not limited to governments, and any actor or group of actors can apply soft powers. For example, they are the principal means of motivation by NGOs that campaign for change using public awareness and/or disruption, and can be the interface between government, the private sector, and NGOs.

A term used by Böcher (2012) for such instruments is cooperative. These measures use the coordination mechanism for negotiations that can lead to voluntary agreements. Examples of such measures are certification schemes and roundtables. Governments can establish cooperative measures but they do not necessarily require government involvement. The understanding of cooperative measures in this study is that the cooperation relates to the other measures in the sector: therefore, the coordination mechanism is used between the measures and not between the actors involved in the cooperative measures themselves. Many instruments that involve public economic cooperation and development agencies, such as the multilateral cooperation with NGOs and international organizations, are included within this type of measure.

### 2.4.3. Voluntary Private Measures

A further means of motivation to change occurs when the private sector applies voluntary private measures in reaction to the same indications of societal demand that give governments their mandate to intervene. The actors in such measures can be private actors from any of the three sectors of the economy, which are the following: the primary sector including e.g., agriculture and mining, the secondary sector including e.g., manufacturing and construction, and the tertiary sector known as the service industry. For reasons of simplicity and coherence, the actor of this type of measure is referred to as "industry" in this study. Industry measures can be either self-regulation or voluntarism. Self-regulation is the case in which industry sectors or an organized group of companies regulate the behavior of its members, which can be done by the implementation of codes of conduct or commitments to a certain objective. Voluntarisms, on the other hand, is the case when individual companies change their behavior. Gunningham et al. (1998) states that self-regulation and voluntarism often include government participation, but in this study voluntary private measures include only those in which no government is involved. Nevertheless, there may be interactions with other measures in which the government is part.

### 2.4.4. Interactions

Earlier approaches to Smart Mixes have primarily focused on the need for combining different approaches to regulation, namely international and national, binding and voluntary. With this definition, we instead focus on the individual parts required for a Smart Mix, which adds nuance and development to earlier discussions. Furthermore, we argue that a truly smart mix is not just a collection of individual measures but is rather an integrated approach in which the different pieces interact with one another so as to collectively improve the results of the whole. The second criterion of Smart Mix identification is to establish the requirement of interaction between the mix of identified measures that are relevant to the geographical and thematic scope of the supply chain. The third criterion is to establish whether interactions between measures lead to improved performance of at least one of the measures, which is expressed in improvements to their capacity to achieve their stated goals. Therefore, the analysis of performance of a Smart Mix does not focus on the general achievement of a sustainable supply chain, but rather on whether the interactions between relevant measures enhance the goal achievement of the measures.

### 3. Methodology

*3.1. Operationalizing the Smart Mix Concept*

The definition of a Smart Mix, along with an understanding of what is meant by the measures that form the mix, is necessary for operationalizing the concept of Smart Mixes. Three fundamental conditions must be met for a mix of measures to be considered a Smart Mix, which are shown in Figure 1.

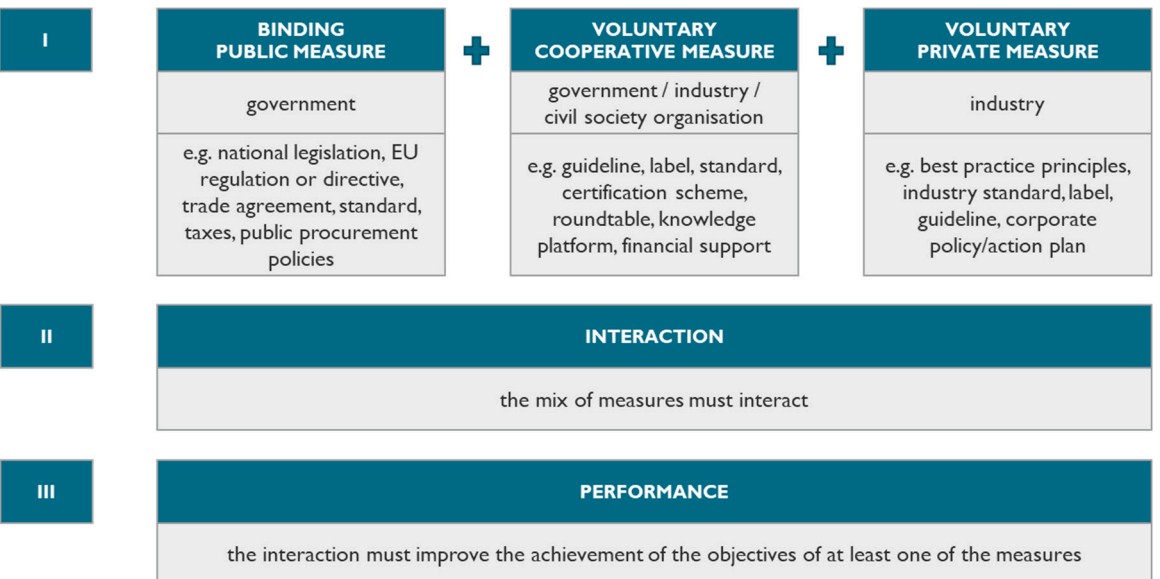

**Figure 1.** Components and conditions of a Smart Mix.

The required measures must (1) be in place and potentially influence the behavior of supply chain actors outside the jurisdiction of the intervening government, (2) they must interact, and (3) the interactions must lead to at least one of the measures better achieving its goals. This suggests that operationalization of the Smart Mix concept is a three-step process.

1.  The first step in the method is to collect and collate the measures that exist within the sector, which includes evaluation of whether the measures influence supply chain actors outside the jurisdiction of the intervening government to change their behavior.
2.  The second step in the method is to evaluate whether the measures interact, which implies gaining a deep understanding of each measure, including their stated goals, and of published documentation including interpretation documents.
3.  The third step is to evaluate whether the interactions between the measures are indeed "smart" and lead to an improvement in performance of at least one of the measures.

We use the example of organic products imported into Switzerland and sold as "certified organic" to demonstrate the proposed operationalization of Smart Mixes using a real-world example. In the organic sector, the consumer relies on cues that signal assurance of integrity (Martinez and Epelbaum 2011), foremost of which are externally issued certificates of organic compliance. Individual producers communicate compliance with standards that are policed by certification bodies, thereby demonstrating the integrity of the certified products to consumers by means of labels, which is analogous to branding. The certification bodies can promote the value of recognition of its label, or brand, as a marketing advantage to producers (Sethuraman and Naidu 2008). Furthermore, certification plays a role along the entire supply chain and is also used by organic producers to identify products that are approved for use in certified production (Fabiansson 2013). However, for certification to prevent fraud, a mechanism is needed to prevent free riders simply printing labels that indicate certification, without diminishing the dynamism in the

sector that Chandler (2006) points out is needed. This mechanism, and how it functions in the Swiss context, is understood as the Smart Mix governing organic products imported into Switzerland.

*3.2. Data Collection and Analysis*

Data were collected by means of desk research, which was supplemented by five systematizing expert interviews. The desk research consisted of identifying and viewing documents that reference to relevant measures, including published material, websites, and interpretation documents, with a particular focus on explicit cross-referencing within the published documents, which was a key indicator of interactions between measures.

The systematizing expert interviews focused on technical knowledge, which relates to highly specific knowledge of a field (Bogner et al. 2018) and process knowledge, which captures knowledge that is based on practical experience and the institutional context of actions (Döringer 2021). The interviews further ensured that no key measures had been omitted and enabled us to gain a deeper understanding of how the measures interact than could be gained from reading publicly available documentation of the measures. The respondents were selected based on their knowledge and experience of the organic sector in Switzerland and in particular with regard to importing local produce. Three of the respondents belong to the Research Institute of Organic Agriculture (FiBL), one to the umbrella body of Swiss organic agriculture (BioSuisse), and the fifth to a major Swiss retailer. They were identified using a snowball sampling strategy with the first respondents chosen based on the existing networks of the research team. Interview partners were presented with the results of the desk research and asked to discuss and elaborate on the outcomes. Their feedback was then used to fine-tune the derived system model, which is shown in Figure 2.

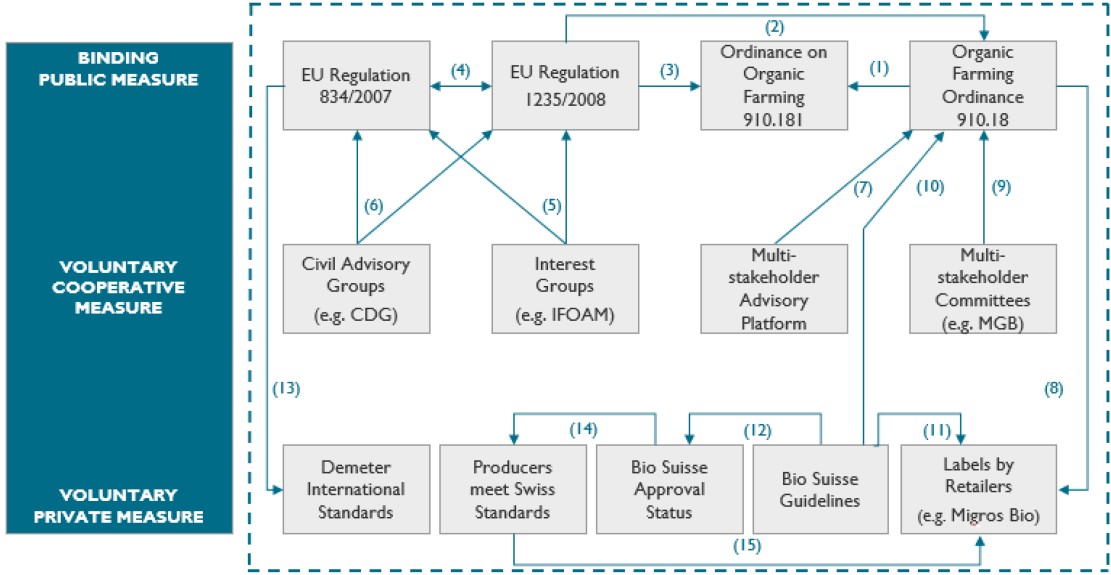

**Figure 2.** Selected key measures in the organic sector and their interactions.

## 4. Results

The key measures identified in the Swiss organic sector, and how they combine and interact to improve performance, are shown in Figure 2, with detailed descriptions of the measures and interactions in Section 4.1. The text in boxes refers to measures that are relevant to the governance of the organic sector in Switzerland in relation to organic imports. The arrows indicate that measures are influenced by other measures. The numbering of these arrows serves to identify the associated explanation in Section 4.1.

*4.1. Smart Mix Criteria 1 and 2: The Mix of Measures and How They Interact*

4.1.1. Binding Public Measures: Organic Regulations

The Swiss Ordinance on Organic Farming and the Labelling of Organically Produced Products and Foodstuffs (Ordinance 910.18) prescribes the standards that must be met for a product, including unprocessed agricultural crop and livestock products, to be considered to have been organically produced and thereby allows products that meet the standards to be labelled as organic (Swiss Federal Council 1997b). Furthermore, the Ordinance describes the inspection procedure along with the sanctions for non-compliance and the requirements for certification bodies. The aim of the Ordinance is to ensure that products in the Swiss market that are labelled as organic have been produced in accordance with the prescribed standards. There is provision in the Ordinance for a list to be created, following the procedure outlined in The Swiss Federal Department of Economic Affairs, Education and Research (EAER) Ordinance on Organic Farming 910.181 (*Interaction 1*), of countries that are able to guarantee that their products meet the requirements. EAER may recognize certification bodies and inspection authorities working in countries that are not included in the list provided the certification bodies and inspection authorities can prove that the products in question meet the requirements. Imported products may be labelled as organic products if they have been produced and prepared in accordance with same rules that apply in Switzerland and have been subjected to an inspection procedure that is equivalent to that in Switzerland. For imports, a certificate of inspection must be issued in the EU system for the electronic certification of the import of organic products (TRACES) in accordance with European Council Regulation (EC) 1235/2008 (*Interaction 2*).

The EAER Ordinance on Organic Farming (Ordinance 910.181) is a companion to Ordinance 910.18 and provides the procedure for creating the list of countries that guarantee that their products meet the Swiss requirements, and outlines further requirements related to gaining the Certificate of Inspection for Imports (Swiss Federal Council 1997a). The aim of the Ordinance (910.181) is essentially to facilitate operationalization of Ordinance 910.18 (*Interaction 1*). Furthermore, it is stated explicitly in this Ordinance (910.181) that the Swiss Federal Office of Agriculture (FOAG) must co-ordinate with the competent authority of the European Commission and inform them of those persons to whom it has issued access rights to the electronic system for exchange of information (TRACES) under EC Regulation No 1235/2008 (*Interaction 3*).

The position of the European Commission, expressed in Council Regulation (EC) No 834/2007 on Organic Production and Labelling of Organic Products (EC 834/2007) is that the organic production method plays a dual societal role (EU Council 2007). Organic production provides for a specific market responding to a consumer demand for organic products and delivers public goods by contributing to rural development, the protection of the environment, and animal welfare. EC 834/2007 defines the objectives, principles, and rules applicable to all stages of production, preparation, and distribution of organic products and their control, and the use of indications referring to organic production in labelling and advertising. The overall aim of EC 834/2007 is to provide the basis for the sustainable development of organic production and to contribute to transparency, consumer confidence, and a harmonized perception of the concept of organic production, while ensuring the effective functioning of the internal market, guaranteeing fair competition, and protecting consumer interests. EC 834/2007 contains provision for the import of products from third countries, as long as the product complies with the requirements for European organic produce. The procedure for establishing equivalence in production and inspection is described in more detail in EC 1235/2008 (*Interaction 4*)

Companion legislation to EC 834/2007 is the Commission Regulation (EC) No 1235/2008 laying down detailed rules for implementation of Council Regulation (EC) No 834/2007 as regards the arrangements for imports of organic products from third countries (EC1235/2008) (European Commission 2008). This includes elaborating on the list of countries in which the production rules and inspection regimes are considered to be equivalent, thereby fulfilling Article 33(1) of Regulation EC 834/2007 (*Interaction 4*). The European Community and the

Swiss Confederation have concluded an Agreement on Trade in Agricultural Products. In order to improve transparency and guarantee the application of EC 1235/2008, an electronic system for exchange of information between the Commission, the Member States, the third countries, and the control bodies and control authorities (TRACES) has been created. The FOAG has a reporting obligation to this system under the Swiss Ordinance 910.18 (*Interaction 3*).

4.1.2. Voluntary Cooperative Measures: Multi-Stakeholder Platforms

Voluntary cooperative measures were identified at both European and Swiss national levels. These were all in the form of multi-stakeholder advisory committees that give advice on the organic standards and their relevant annexes.

The International Federation of Organic Agriculture Movements EU Regional Group (IFOAM EU), which is the European umbrella organization for organic food and farming, provides advice on many areas of organic production to European Commission Institutions (European Commission 2020a). IFOAM EU members include farmer organizations, NGOs, research institutes, certification bodies, and the private sector (IFOAM EU 2020). IFOAM EU and the European Commission collaborate to provide advice to ensure that the EU keeps up to date with any technical changes in the rapidly advancing and highly innovative organics sector (LobbyFacts 2020). Such advice is used to amend the technical annexes of regulations EC 834/2007 and EC 1235/2008 (*Interaction 5*). A recent example is the classification of new CRISPR/Cas breeding processes as genetic engineering under the law.

A similar group, although not of technical experts, is the Civil Dialogue Group (CDG), which comprises representatives of groups including NGOs and environmental charities, producer organizations and cooperatives, industry associations, and trade unions (European Commission 2020b). The CDG's aims are to help to advise and monitor the Commission's organic policy and make contributions based on their on-the-ground experience. To achieve these aims, the CDG assists the Commission in maintaining a regular dialogue on all matters related to organic farming and advises the Commission on relevant policy. In this way, the CDG can also influence the technical annexes of the regulation EC 834/2007 and EC 1235/2008 (*Interaction 6*).

At the Swiss level, Bio Suisse collaborates with the Swiss Farmers Union and the Swiss Agricultural Alliance to create a multi-stakeholder advisory platform for a range of issues related to organic production including providing advice to the Swiss Government, which is used to amend and update the Swiss Organic Farming Ordinance 910.18 (*Interaction 7*). The Swiss Federal Council recognizes that organic farming represents an overall farm production system that makes a major contribution to the preservation of natural resources, soil fertility and biodiversity, so the advice from the multi-stakeholder platform fits within its overall strategy (Swiss Federal Council 2014). Furthermore, the advice given by the platform responds to pressure on the organic sector to renew and innovate against the background of increasing competition from regional and other sustainability labels (Bio Suisse 2018), steadily rising supply of imported organic products, and an increasing number of cases of fraud (Swiss Federal Council 2014).

Also in Switzerland, Migros Bio (MGB) is the private organic label of Migros, the biggest retail company in the country. Businesses that intend to provide Migros with organic products must sign an agreement with the Migros cooperative association to comply with the Migros guidelines, which regulate the requirements for organic food marketed under the label "Migros Bio." The Swiss legal provisions for the production, processing, and marketing of organic food (Swiss Ordinance 910.18) are mandatory and form the basis of the MGB guidelines (*Interaction 8*). The MGB, however, reserves the right to amend their own guidelines provided such amendments are consistent with the Ordinance. All changes are coordinated with organic practitioners, such as organic associations/federations and organic certification bodies, and are examined by the MGB committee quality assurance expert team before being approved by the Migros organic core team (*Interaction 9*).

### 4.1.3. Voluntary Private Measures: Organic Standards

The Association of Swiss Organic Farming Organizations: Bio Suisse, has produced guidelines for the production, processing, and trade of plant and animal products that are marketed with the "bud," which is the Bio Suisse trademark for produce that is certified organic. The bud is the most important private seal for organic products in Switzerland and companies that use it on their products must sign a license agreement with Bio Suisse that guarantees that they comply with the Bio Suisse Guidelines. The Bio Suisse guidelines are based on the Swiss Organic Farming Ordinance 910.18 (*Interaction 10*), but go beyond it in many respects, such as plant protection, animal feed, processing, and social standards. A further major retailer in Switzerland is Coop, who have their own organic label, '"Coop Naturaplan," which explicitly follows the guidelines of the Bio Suisse bud, so the Bio Suisse requirements for imported organic produce also apply to Naturaplan (*Interaction 11*).

The core task of Bio Suisse is to promote Swiss domestic organic producers and products but can contribute to the promotion of organic farming worldwide through imports (Bio Suisse 2020). To this end, Bio Suisse has created an instrument called "approval status of imported products," and thereby allow use of the bud label on imported produce, if domestic production is not possible (such as coffee) or is not sufficient from Swiss bud farmers (such as cereals). The basic prerequisite for the award of the bud label to imported products is that the food is produced abroad in strict accordance with the Bio Suisse guidelines and that the farms are inspected and certified accordingly (*Interaction 12*). A similar case of a farmers' organization creating standards for imported products is Demeter: the brand for products from biodynamic agriculture, which has created the Demeter Production and Processing International Standard for the use and certification of Demeter, Biodynamic, and related trademarks (Demeter-International e.V. 2019). In order to be allowed to use the Demeter trademark on its products imported into Switzerland, a company needs to sign an agreement with Demeter Switzerland. The Demeter International Standard directly refers to EC regulation 834/2007 (*Interaction 13*).

The price of vegetables in Switzerland is 40% higher than the average European market price and is even higher for organic produce (Chevalley 2018), so farmers outside Switzerland seek to access the lucrative Swiss organic market by producing in accordance with Ordinance 910.18. However, the organic price premium is only available to certified and labelled produce that meet the Swiss standards including the inspection routine. In this way, farmers outside the Swiss jurisdiction voluntarily follow Swiss law and comply with the requirements of the Bio Suisse bud (*Interaction 14*). The producers can then enter supplier agreements with the major retailers (*Interaction 15*).

### 4.1.4. Criterion 3: Performance

It is a stated goal of Bio Suisse to contribute to the promotion of organic farming worldwide by promoting imports as long as the imports complement, rather than compete, with the domestic bud range (Bio Suisse 2018). Organic farmers who wish to export produce to Switzerland have an interest in gaining the price premium as a reward for the extra costs involved with organic production, while the producing farmer must also be protected from unfair competition from fraudulent products being sold as organic. The ability of these farmers to gain approval status from Bio Suisse enables them to achieve the price premium while simultaneously protecting them from fraudulent competitors who claim to have produced organically. The approval status also enables the farmers to enter supply agreements with the private labels held by the major Swiss retailers: Migros and Coop. In this way, the Swiss producer organizations, supported by the legislation, can persuade farmers outside the Swiss jurisdiction to follow Swiss standards and inspection routines voluntarily, in exchange for access to the lucrative Swiss market.

The interactions between the advisory committees at the EU level, interest groups and civil advisory groups, and at the Swiss level, the multi-stakeholder advisory platform and the multi-stakeholder committees, contribute to ensuring that the regulations and ordinances remain relevant and effective by continually adapting to the dynamism in the

sector. In this way, these groups directly contribute to the regulatory goals and thereby add value to the supply chain. The regulations and ordinances in turn, protect the private industry standards by preventing use of the protected terms, such as "organic," unless the requirements described in the standards have been met. These interactions are a clear synergy between the interests of the private sector in Switzerland and the European and Swiss legislating bodies that encourage the dual societal role played by organic production of providing for the consumer demand for trustworthy organic products and delivering environmental and social public goods.

## 5. Discussion

Kinderman's (2016) doubts about the practicality of the Smart Mix concept were based around questions about whether businesses will engage with measures that they perceive to be against their interests. While this may be the case when hard powers are applied to coerce industry to converge with public policy, the Swiss organic sector is an example in which legislation gives voluntary measures, which were created by industry, protection from free riders and thereby aligns the interests of government and the private sector. The development of the Smart Mixes in the Swiss organic import case is compatible with the conclusions of Amstutz and Karavas (2009) who argued that incremental, stakeholder-dialogue-based development of legislation/regulation might be a typical approach in transnational and EU contexts. Although Amstutz and Karavas (2009) see that such means of legislating would be prevalent in transnational contexts where there is no single clearly authoritative legislator, such dialogue was also evident in the Swiss organic imports case, so may be relevant to other national legislation in which stakeholders and interest groups are increasingly integrated into decision-making. However, we should remember that the case study example is further complicated by the far-ranging integration of Swiss legislation into the EU framework. Furthermore, the development is compatible with recent work on public-private governance interactions, such as (Renckens 2020) and with the explanation of Arcuri (2015) in that the case represents the publicization of private governance.

The problematic power asymmetries along and surrounding global business relations, with a dominance by the "buying" country, have been quite well documented in studies on global value chains since they were pointed out by Kindleberger (1981). Indeed, many businesses have outsourced their production to poorer developing countries with low salaries and lax social and environmental legislation because of efforts to reduce the cost of production (Cossart et al. 2017) and certification tools, such as organic, have tended to not address the structural roots of these problems (Guthman 2014). However, this is a criticism of the organic standards rather than of the organic certification system *per se*, and cannot be understood as a suggestion that consumer countries should take no action. Our argument is to the contrary: that consumer countries have a moral imperative to take action to address such imbalances, and we propose the working definition of Smart Mixes as guidance for designing mixes of measures that may most effectively do so.

Rees' (2019, n.p.) assertion that a true Smart Mix means looking beyond the measures that are most convenient, or already in place, implies that positive government intervention, such as creating and implementing legislation or establishing multi-stakeholder platforms, should at least be considered to complete an existing mix of measures in a given sector so that they interact and enhance the achievement of goals. The Smart Mix definition presented in this study, which readily provides criteria for identifying whether a Smart Mix exists, can also be used for the converse: to identify which criteria must be filled for a Smart Mix to exist. In this way, the definition serves to identify which elements of a Smart Mix are missing and thereby lead to informing which remedial actions might be taken.

Several pieces of legislation were identified at both EU and national levels, which were found to provide legislative support to organic standards that had been developed by the private sector. These pieces of legislation provide the starting point for what Kinderman (2016, p. 32) calls "Regulated Private Governance (RPG) arrangements" that have the characteristics of private measures. Furthermore, multi-stakeholder platforms provide policy

advice that is reflected in updating and amending the supporting legislation and organic standards. The standards combine to protect organic producers from unfair competition from producers seeking to gain the price premium without fulfilling the requirements outlined in the standards. The combination of these results allows the conclusion the Smart Mix concept is valid, at least under certain conditions, and is a useful approach to examining the governance structures of a sector that operates in multiple jurisdictions

This study is subject to two main limitations. Firstly, it is an industry- and country-specific study and, therefore, more cross-sectoral and cross-national research is required to test the reliability of this smart-mix focus. Amstutz and Karavas (2009) suggest that the kinds of Smart Mixes proposed in this contribution are in many ways already a standard means of legislating in specific contexts, such as in the EU, but perhaps not in others. However, given that there are typically extra costs associated with sustainable production, it is reasonable to assume that the Smart Mix concept might be equally applicable to other supply chains with extra costs and associated price premiums. Further research would be advised to examine the Smart Mix concept in other supply chain contexts to gain further insight into its generalizability.

The second main limitation is that the focus was solely on an analysis of Smart Mixes as an approach to influencing the behavior of supply chain actors, with less attention given to alternative approaches, such as with due diligence legislation. Although mandatory due diligence would ease the certification burden for producers, who are often in the global South, and the approach is increasingly proposed as a solution for supply chain sustainability, its potential in the organic sector has been insufficiently explored. There is, however, evidence from other sectors that mandatory due diligence may provide inadequate assurance to consumers due to the difficulties of importers in evaluating the production processes abroad. On the other hand, mandatory due diligence may have the potential to complement the identified Smart Mix in the organic sector, and may even provide a means of addressing the weaknesses of organic certification in addressing social issues that were identified by Guthman (2014). Future research might also investigate whether a Smart Mix was indeed necessary in the organic sector or whether similar outcomes may have been achieved by other means.

## 6. Conclusions

The aim of this study was to create a working definition of a Smart Mix, which would establish the Smart Mix concept as a viable approach with which governments of consumer countries can motivate or cooperate with industry and thereby contribute to the sustainability of the supply chains of consumer products. To address this aim, we derived a working definition of a Smart Mix from existing literature and illustrated it by applying it to the organic produce sector in Switzerland. Given that the definition required examination of measures that influence behavior outside the jurisdiction of the legislating body, we focused on imports of organic produce. The Smart Mix in the Swiss organic sector essentially comes down to cooperation between public and private actors, with a focus on binding public measures providing protection for the private sector. The potential for Smart Mixes appears to lie in convincing industry that adopting voluntary measures to satisfy societal demand is a less odious path than resistance to regulation. Although this is hardly evidence that co-governance is truly a source of social innovation, it is consistent with Ruggie's (2011) conception of a Smart Mix in which voluntary industry efforts would combine with legal developments to produce sustainability outcomes (Wettstein 2015).

The working definition enabled the analysis of a Smart Mix that starts from public demand for sustainably produced goods, which gives industry a motivation to participate in voluntary measures while giving governments a mandate to adopt the position as policy to protect the sector with legislation. This policy position should then lead to government actions that lead to collaboration, rather than confrontation, with the private sector. Although this path sounds overly optimistic, this research found such a convergence in the organic sector in Switzerland and it is conceivable that such convergence exists

in further sectors. In any case, the creation of the working definition and analytical process described in this paper presents future researchers and/or administrators who are interested in evaluating the governance of any international supply chain with a useful analytical tool for identifying the adequacy of existing measures and how they interact.

**Author Contributions:** Conceptualization, R.H., M.W. and C.S.; methodology, R.H., M.W. and C.S.; validation, R.H., M.W. and C.S.; formal analysis, R.H. and M.W.; investigation, R.H. and M.W.; resources, C.S.; data curation, R.H. and M.W.; writing—original draft preparation, R.H. and M.W.; writing—review and editing, C.S.; visualization, M.W.; supervision, C.S.; project administration, C.S. and M.W.; funding acquisition, C.S. All authors have read and agreed to the published version of the manuscript.

**Funding:** The literature review part of this submission was co-funded by the German Federal Ministry for Economic Cooperation and Development (BMZ) and cofunded by the Swiss National Science Foundation (SNSF) through the project "Enhancing supply chain stability, resilience and sustainability through improved sub-supplier management –chocolate and cotton apparel case studies" within the National Research Programme 73 "Sustainable Economy". The views and opinions expressed herein are those of the authors and do not necessarily reflect those of the BMZ or SNSF. The designations and terminology employed may not conform to BMZ or SNSF practice and do not imply the expression of any opinion whatsoever on the part of the BMZ or SNSF. Neither SNSF nor BMZ are liable for any use that may be made of the information contained herein. The primary research in this submission received no external funding.

**Data Availability Statement:** The data used in this study are considered to be personal data in accordance with the European General Data Protection Regulations (GDPR). The qualitative nature of the data means that it cannot be effectively anonymised and therefore may not be made publicly available in accordance with the GDPR (Article 26). Data will be made available from the authors on request under the condition that both request and supply are GDPR compliant.

**Conflicts of Interest:** The authors declare no conflict of interest.

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
