# Peer review of "Smart Mixes in International Supply Chains: A Definition and Analytical Tool, Illustrated with the Example of Organic Imports into Switzerland"

_admsci, doi:10.3390/admsci11030099_

Round 1

Reviewer 1 Report

The manuscript submitted for review presents a research problem that is important from the scientific point of view. A properly developed Smart Mixes concept and its practical applications can significantly affect the continuity of the supply chain. It is especially true during emergencies such as the current COVID-19 pandemic. The discussed example of Switzerland and the aspect of sustainable development are definitely related to ecology, which may be additionally interesting for potential readers.
The layout of the manuscript is correct and includes the elements required for such works.
Because of the above, I believe that the manuscript is likely to arouse interest and is suitable for printing. However, subject to the following corrections by the authors:
1. The "Abstract" section should not contain references to literature (Ruggie, 2011)
2. The manuscript is based on literature analysis. Still, unfortunately, it lacks the methodological part, in which the authors should describe how they selected the sources, what methods they used, why, etc. I propose to distinguish a part of "Materials and Methods" or "Methodology" where filled.
3. DOI is missing in many places in the References section.

Author Response

Response 1): The reference has been removed in abstract.

Response 2): A methodology section has been added, which describes the process for using the newly derived definition of Smart Mixes to analyze international supply chains.

Response 3): These DOI have been added in the reference section.

Reviewer 2 Report

After re-reading the piece, you are really talking about operationalization and conceptualizing smart mixes. But your total refers only to "definitions" which is "conceptualizing." They go hand in glove, and not to make a long title even longer, I'm wondering if both oper/concept. could be worked into the title at some point?

Also, this is really a case study. The reader is lured here thinking s/he will be reading about SM's globally, but alas, this is not so. Is this bait and switch? Should Switzerland be in the title? If not, then you need to add some examples that are non-Swiss. I've pointed out some instances below.

Some minor points. Mix up sentence starters with "However" line 119 and then further below. 

In the literature review, I'm struck by the absence of references to MITI in Japan in the post WW II era. The ministry plaid a key role in harveting the private sector and offering generous subsidizies to build up what turned out to be world-class industrial and chemical powerhouses.

In the same light, the chabols of South Korea pose an alternative model, largely private sector ones. However, their impact has been undeniable. Now, I'm not suggesting they are Smart Candidates 100% but they do reflect a kind of public-private partnership of sorts. Alas, is your paper really about something new or is this old wine in a new bottle. I would like to see at least some mention of this "collaboration" (for lack of a better term).

Can Table 1 be used without the word "mix" in the definition. Rule number one in terms: Never use the word (or a root-word derivative) in the definitition itself. Thus, synonyms like "blend", "combination," "Inputs" "derivations" etc. make for impactful writing. 

Tabel 2. Reassess  your 1 + 2+ 3 columns because "certification" is hardly ever a mandate (it may be in Switzerland). Thinking of all the agricultural and organic farming "certifications" which exist in North America and note that affiliation is all voluntary. In Table 2, your certification scheme falls under binding measures. Reexamine all categories closely, please. Are these Swiss specific only? Can you give Swiss, EU, Asian and North American examples elsewhere in the Table to illustrate? Remember: all your readers will not be Swiss. 

Need more industry-specific examples like EUTR on line 291 to flesh out your examples, and the more you bring into place outside of Switzerland, the better. Perhaps some mention of the EU should be in the title. The paper needs a geographic anchor/descriptor since it is pitched globally, but is not really.  The scholar in Mexico and Nigeria need to take away lessons this we need apple to apple comparisons. 

Ah hah! Your figure 2 confirms my sentiment all along, something not reflected in the title. This is all about Switzerland. Thus it is incumbent upon you to work Switzerland into the title; your task then is to try to generalize to other nations. Otherwise, this piece belongs in a different journal.

When you write: "The key measures identified 399 in the Swiss organic sector, and how they combine and interact to improve performance, 400 are shown in Figure 2. " The reader sees only arrows going in one direction. How can there exist interaction and improvement if a change is unidirectional? And, sorry, but this Figure is too hard to interpret. What are the small blue-lettered numbers indicating? Stages? Steps in a process? You need more attention to this model; e.g., the text does not address and this reader cannot really follow the logic. How will you defend yourselves from the criticism that your Smart Mixes definition is mechanistic and linear (in its proposal, and not in real life)?

The next few pages that follow from here on the organic sector really bogged this reader down in too much detail. The non-Swiss reader wants to be abl to see lessons learned from the Swiss experience and not sit for an examination of Swiss/EU agricultural organic certification procedures. This part needs to be greatly reduced, greatly. Summarize the key points with much less text and some sort of filtering process; tell the reader how you make sausage out this smart mix. Do it simply and eloquently. 

Am wondering if the term "Bio Suisse" needs to be in the title at this point? 

If this is what your paper is really about "These interactions are a clear syn- 570 ergy between the interests of the private sector in Switzerland and the European and Swiss 571 legislating bodies that encourage the dual societal role played by organic production of 572 providing for the consumer demand for trustworthy organic products and delivering en- 573 vironmental and social public goods. "  --- then the title needs to reflect this (as does the abstract and conclusion). Accordingly, you cut out everything this is not taking your readers through this (rather long) discussion. All is becomes superflous. Edit, refocus, redefine.

Let me return to the old wine/new bottle metaphor. When I finished the piece it struck me that you were really talking about a policy context that encourages a firm or sector to engage in doing what is correct both ethic ally and economically (hopefully). This is really CSR but the term corporate social responsibility only appears on pp. 2 and 3, in separate forms. This raises even more doubts about your contribution to the literature might be. 

"Smart Mix Lessons from the Swiss Organic Sector" might be a more accurate title. Also, because the Swiss have some of the highest consumer purchasing power in the world, will consumers elsewhere be able to take away lessons learned here? Who benefits from understanding your explanation of the global supply/value chain? 

In your conclusions, you note "Several pieces of legislation were identified at both EU and national levels, which 629"  which reinforces the idea that this paper is very EU focused and thus this fact needs to be reflected in your title. As it stands, the reader believes to be entering a global garden when alas it is merely a Swiss/European one. Fairness in advertising/labeling should preval in the title. 

I don't recall seeing any consumer survey evidence in the paper to support his conclusion: "The measures combine to enhance public confidence in the authentic- 635 ity of organic produce, while protecting organic producers from unfair competition from 636 producers seeking to gain the price premium without fulfilling the requirements outlined 637 in the standards. "   [you'll need lots of data to corroborate this finding]

You talk about the burden placed on the Global South in subscribing to this normative Smart Mix, and how often this can be onerous (think: coffee, bananas, cacao). However, nowhere in the paper is Nesltlé mentioned! This seems to be a major oversight: a paper on smart mix that uses the organic sector in Switzerland and does not mention one of the largest world producers!

Your last sentence ---which will be read often-- strikes me as a bit feckless: "Although this path sounds overly optimistic, this research 674 found such a convergence in the organic sector in Switzerland and it is conceivable that 675 such convergence exists in further sectors. "

You've asked the reader to climb a high Swiss mountain, and then you simply say: This is our model, here it is, take it. I'm not sure that's fair wihtout a clear way being outlined by you that might point out how that convergence might come about. Therefore I propose that half of the paper focus on the Swiss sector an the other half (one quarter each) on how other nations have sucessfully and unsucessfully forged their own Smart Mix. Keep it it in the organic ag realm to make comparisons apples to apples. If you can do this, you will have a more valuable contribution. As it stands now, this is a nicely-researched single-sector and single-nation case study. And I'm not sure that smart mix is not old wine in a new bottle. I urge you to convince me otherwise.

Thank you and good luck in revising this paper. 

Author Response

Reviewer 2

R2: After re-reading the piece, you are really talking about operationalization and conceptualizing smart mixes. But your total refers only to "definitions" which is "conceptualizing." They go hand in glove, and not to make a long title even longer, I'm wondering if both oper/concept. could be worked into the title at some point?

R2: Also, this is really a case study. The reader is lured here thinking s/he will be reading about SM's globally, but alas, this is not so. Is this bait and switch? Should Switzerland be in the title? If not, then you need to add some examples that are non-Swiss. I've pointed out some instances below.

R2: In the literature review, I'm struck by the absence of references to MITI in Japan in the post WW II era. The ministry plaid a key role in harveting the private sector and offering generous subsidizies to build up what turned out to be world-class industrial and chemical powerhouses.

In the same light, the chabols of South Korea pose an alternative model, largely private sector ones. However, their impact has been undeniable. Now, I'm not suggesting they are Smart Candidates 100% but they do reflect a kind of public-private partnership of sorts. Alas, is your paper really about something new or is this old wine in a new bottle. I would like to see at least some mention of this "collaboration" (for lack of a better term).

Response: We are aware that PPPs are an effective way of creating economic and industrial advantage. However, our paper is about what governments can do to convince the private sector to adopt responsibility for factors that are traditionally considered to be public goods, such as protection of the environment and of human rights. We feel that to include examples of PPPs that are not related to public goods would confuse, rather than clarify, the main points of the paper. The revision of the literature review, and the addition of the subheadings, has made the focus of the literature review clearer.

R2: Some minor points. Mix up sentence starters with "However" line 119 and then further below.

Response: The sentence has been revised.

R2: Can Table 1 be used without the word "mix" in the definition. Rule number one in terms: Never use the word (or a root-word derivative) in the definitition itself. Thus, synonyms like "blend", "combination," "Inputs" "derivations" etc. make for impactful writing. 

R2: Need more industry-specific examples like EUTR on line 291 to flesh out your examples, and the more you bring into place outside of Switzerland, the better. Perhaps some mention of the EU should be in the title. The paper needs a geographic anchor/descriptor since it is pitched globally, but is not really.  The scholar in Mexico and Nigeria need to take away lessons this we need apple to apple comparisons. 

R2: Tabel 2. Reassess  your 1 + 2+ 3 columns because "certification" is hardly ever a mandate (it may be in Switzerland). Thinking of all the agricultural and organic farming "certifications" which exist in North America and note that affiliation is all voluntary. In Table 2, your certification scheme falls under binding measures. Reexamine all categories closely, please. Are these Swiss specific only? Can you give Swiss, EU, Asian and North American examples elsewhere in the Table to illustrate? Remember: all your readers will not be Swiss. 

Response: The examples are only included to illustrate what is meant and are not Switzerland specific. The clarification of what the measures include is expanded in much more detail under the heading: “Clarifying key terms” above. Although there are many examples of certificates that are mandated (think fire safety certificates for public buildings for example), we have removed it from the figure to avoid misunderstanding.

R2: When you write: "The key measures identified 399 in the Swiss organic sector, and how they combine and interact to improve performance, 400 are shown in Figure 2. " The reader sees only arrows going in one direction. How can there exist interaction and improvement if a change is unidirectional? And, sorry, but this Figure is too hard to interpret. What are the small blue-lettered numbers indicating? Stages? Steps in a process? You need more attention to this model; e.g., the text does not address and this reader cannot really follow the logic. How will you defend yourselves from the criticism that your Smart Mixes definition is mechanistic and linear (in its proposal, and not in real life)?

Response; These points are addressed in section 4.1, which describes the interactions in detail. The figure is indeed complex, which is why section 4.1 takes around two pages to explain it. There is another comment asking us to reduce the explanation, but we feel that the explanation is needed. Instead, we have added a paragraph to indicate to the reader that section 4.1 is the explanatory text to Figure 2.

R2: Ah hah! Your figure 2 confirms my sentiment all along, something not reflected in the title. This is all about Switzerland. Thus it is incumbent upon you to work Switzerland into the title; your task then is to try to generalize to other nations. Otherwise, this piece belongs in a different journal.

R2: The next few pages that follow from here on the organic sector really bogged this reader down in too much detail. The non-Swiss reader wants to be abl to see lessons learned from the Swiss experience and not sit for an examination of Swiss/EU agricultural organic certification procedures. This part needs to be greatly reduced, greatly. Summarize the key points with much less text and some sort of filtering process; tell the reader how you make sausage out this smart mix. Do it simply and eloquently. 

R2: Am wondering if the term "Bio Suisse" needs to be in the title at this point? 

Response: In response to other comments, there is less emphasis on the role of Bio Suisse in the illustration of Smart Mixes, so we have left it out of the title.

R2: If this is what your paper is really about "These interactions are a clear syn- 570 ergy between the interests of the private sector in Switzerland and the European and Swiss 571 legislating bodies that encourage the dual societal role played by organic production of 572 providing for the consumer demand for trustworthy organic products and delivering en- 573 vironmental and social public goods. "  --- then the title needs to reflect this (as does the abstract and conclusion). Accordingly, you cut out everything this is not taking your readers through this (rather long) discussion. All is becomes superflous. Edit, refocus, redefine.

R2: Let me return to the old wine/new bottle metaphor. When I finished the piece it struck me that you were really talking about a policy context that encourages a firm or sector to engage in doing what is correct both ethically and economically (hopefully). This is really CSR but the term corporate social responsibility only appears on pp. 2 and 3, in separate forms. This raises even more doubts about your contribution to the literature might be. 

R2: I don't recall seeing any consumer survey evidence in the paper to support his conclusion: "The measures combine to enhance public confidence in the authentic- 635 ity of organic produce, while protecting organic producers from unfair competition from 636 producers seeking to gain the price premium without fulfilling the requirements outlined 637 in the standards. "   [you'll need lots of data to corroborate this finding]

Response: The reference to consumers has been removed.

R2: You've asked the reader to climb a high Swiss mountain, and then you simply say: This is our model, here it is, take it. I'm not sure that's fair wihtout a clear way being outlined by you that might point out how that convergence might come about. Therefore I propose that half of the paper focus on the Swiss sector an the other half (one quarter each) on how other nations have sucessfully and unsucessfully forged their own Smart Mix. Keep it it in the organic ag realm to make comparisons apples to apples. If you can do this, you will have a more valuable contribution. As it stands now, this is a nicely-researched single-sector and single-nation case study. And I'm not sure that smart mix is not old wine in a new bottle. I urge you to convince me otherwise.

Response: We are indeed asking the reader to climb a high mountain as this way of thinking about international supply chains has not had a lot of attention in the literature. But we are not asking anybody to compare anything (yet). We expect to do so in the future as we apply the analytical approach to other sectors, but for now, we are presenting an analytical approach, and demonstrating that it works using the real-world example of a sector in which a Smart Mix has been implemented. A result that some academics in the past have expressed as being unrealistic.

R2: You talk about the burden placed on the Global South in subscribing to this normative Smart Mix, and how often this can be onerous (think: coffee, bananas, cacao). However, nowhere in the paper is Nesltlé mentioned! This seems to be a major oversight: a paper on smart mix that uses the organic sector in Switzerland and does not mention one of the largest world producers!

Response: A Smart Mix should ease the burden of the global south. Not add to it. It’s about the consumer countries adopting a share of the responsibility for sustainability. There are some companies that we know of that are part of Smart Mixes, but Nestlé, along with very many other companies, is not one of them.

R2: "Smart Mix Lessons from the Swiss Organic Sector" might be a more accurate title. Also, because the Swiss have some of the highest consumer purchasing power in the world, will consumers elsewhere be able to take away lessons learned here? Who benefits from understanding your explanation of the global supply/value chain? 

Response: The title has been changed as requested. The concept of Smart Mixes is not aimed at consumers but is rather intended for the academic and administrative world, and in particular those who wish to analyze the governance of an international supply chain. We have made that clearer in the revised text.

Reviewer 3 Report

Interesting paper and good materials. However, I believe authors need to make a major revision to publish this paper.

Abstract:

Please remove references from the abstract.

The abstract needs to be re-written and rigorously developed in line with manuscript results. It is not clear about the originality of the study, the gap, what you have done in this paper, what are the result.

Introduction:

Introduction part should re-written and reformulated more logically. You need to start with general explanation of the problem and then narrow down to the problem statement. Start with the clarification of Smart Mixes, then how it can solve problem. What is the problem in this paper, how Smart Mixes can help to solve this problem? Then, providing some evidence that this issue hasn’t bee addressed in the literature (gap), and then your contribution. Finally last paragraph should be about the structure of the paper

Literature Review:

The flow in the literature review is not clear. I suggest making the logical flow clear between sections and paragraphs in the literature review.

Solution Methodology:

I didn’t see any sing of methodology. The authors should make it clear which methodology they used to develop this paper. otherwise, there is a big concern in this paper.

Results and Discussion:

The discussion should be more stronger answering the following questions:

How the findings of this study stand against findings from other similar research? What is the novelty of the research works? What s the implications for mangers and also contributions for literature. It is not clear.

Conclusion:

Conclusion should include limitations and future directions. It is not clear now.

Author Response

R3: Please remove references from the abstract.

Response: The reference has been removed.

Response: The abstract has been rewritten according to the format suggested by the reviewer.

R3: Introduction part should re-written and reformulated more logically. You need to start with general explanation of the problem and then narrow down to the problem statement. Start with the clarification of Smart Mixes, then how it can solve problem. What is the problem in this paper, how Smart Mixes can help to solve this problem? Then, providing some evidence that this issue hasn’t bee addressed in the literature (gap), and then your contribution. Finally last paragraph should be about the structure of the paper

R3: The flow in the literature review is not clear. I suggest making the logical flow clear between sections and paragraphs in the literature review.

R3: I didn’t see any sing of methodology. The authors should make it clear which methodology they used to develop this paper. otherwise, there is a big concern in this paper.

Response: A methodology section has been added.

R3: The discussion should be more stronger answering the following questions:

How the findings of this study stand against findings from other similar research? What is the novelty of the research works? What s the implications for mangers and also contributions for literature. It is not clear.

R3: Conclusion should include limitations and future directions. It is not clear now.

Response: Limitations and future research have been added in the discussion section. We feel that this new text fits better with the narrative there.

Round 2

Reviewer 1 Report

Good work.

Author Response

Thanks, we very much appreciate the thoughtful and constructive criticism that enabled us to improve the paper.

Reviewer 2 Report

This is a  nice revision. I would only ask that the authors include one or two sentences pointing to the possible limitations of the study: That it is an industry- and country-specific study and, therefore, more cross-sectoral and cross-national research is required to test the reliability of this smart-mix focus. 

Author Response

Thank you. We very much appreciate the considered and constructive comments that have helped us to improve the paper.

Response: We have added some new text at the end of the discussion section. The new text reads: “This study is subject to two main limitations. Firstly, it is an industry- and country-specific study and, therefore, more cross-sectoral and cross-national research is required to test the reliability of this smart-mix focus. […..] Further research would be advised to examine the Smart Mix concept in other supply chain contexts to gain further insight into its generalizability.”

And for the second limitation, we have added the text: “The second main limitation is that the focus was solely on an analysis of Smart Mixes as an approach to influencing the behavior of supply chain actors, with less attention given to alternative approaches, such as with due diligence legislation.[…] Future research might also investigate whether a Smart Mix was indeed necessary in the organic sector or whether similar outcomes may have been achieved by other means.”

Reviewer 3 Report

The logical flow in the paper should be improved, and the methodology section could be better presented.

Author Response

Thank you for the constructive criticism.

Reviewer 3:  The logical flow in the paper should be improved.

Response: We believe that this comment has been addressed by the revisions to the draft.  Firstly we write about governance of supply chains and the imperative to consider sustainability before outlining the problem of governments in consumer countries being limited by jurisdictional constraints.  We then discuss the Smart Mix concept, which has been proposed as a solution, and point out that its implementation has been hindered by a lack of agreement in what a Smart Mix actually is (lack of definition). We then go to the literature and derive a definition along with some steps as to how this might be implemented, and define the key terms of the definition. To illustrate how the concept might be implemented, we apply the derived steps to the real world example of organic imports into Switzerland.

To highlight this flow, we have revised the text with a focus on logical flow, revised the abstract, added the new 'structure' paragraph at the end of the introduction section to outline the structure of the arguments that follow, and inserted numbered sub-headings that function as signposts for the reader.

We are however open to making amendments should there be further ways for us to make the logical flow clearer.

Reviewer 3:  The methodology section could be better presented.

Response: The literature review now includes a description of the analytical framework, including an outline of the steps in operationalizing a Smart Mix. We believe this is sufficient for future scholars to replicate the study. Indeed, presentation of this method for analyzing a supply chain to identify a Smart Mix is a key message of the paper.

In the methods section, we have explained the selection of the case study and provided the theoretical justification of the expert interview approach, before describing the sampling method and the procedure. We are however open to making amendments should there be further ways for us to improve the presentation of the methodology section.